# Essential Oils from Two Apiaceae Species as Potential Agents in Organic Crops Protection

**DOI:** 10.3390/antibiotics10060636

**Published:** 2021-05-26

**Authors:** Nadjiya Merad, Vanessa Andreu, Slimane Chaib, Ronaldo de Carvalho Augusto, David Duval, Cédric Bertrand, Yacine Boumghar, André Pichette, Nassim Djabou

**Affiliations:** 1Laboratoire COSNA, Département de Chimie, Faculté des Sciences, BP 119, Université de Tlemcen, Tlemcen 13000, Algeria; ninaruby-tlem@hotmail.fr; 2AKINAO, 52 Avenue. Paul Alduy, 66860 Perpignan, France; vanessa.andreu@akinao-lab.com (V.A.); cedric.bertrand@univ-perp.fr (C.B.); 3Université de Perpignan, PSL Research University: EPHE-UPVD-CNRS, USR 3278 CRIOBE, 52 Avenue Paul Alduy, CEDEX, 66860 Perpignan, France; slimane.chaib@univ-perp.fr; 4Université Perpignan Via Domitia, 52 Avenue Paul Alduy, CEDEX, 66860 Perpignan, France; ronaldodecarvalhoaugusto@gmail.com (R.d.C.A.); david.duval@univ-perp.fr (D.D.); 5Laboratoire de Biologie et Modélisation de la Cellule, CNRS, Université Lyon, 69007 Lyon, France; 6CÉPROCQ, College of Maisonneuve, 6220 Rue Sherbrooke est, Montréal, QC H1N 1C1, Canada; 7Centre de Recherche sur la Boréalie (CREB), Laboratoire LASEVE, Université du Québec à Chicoutimi (UQAC), 555, Boulevard de l’Université, Chicoutimi, QC G7H 2B1, Canada; Andre_Pichette@uqac.ca

**Keywords:** *Eryngium triquetrum*, *Smyrnium olusatrum*, essential oils, chemical composition, crop protection, falcarinol

## Abstract

Chemical composition and herbicidal, antifungal, antibacterial and molluscicidal activities of essential oils from Choukzerk, *Eryngium triquetrum*, and Alexander, *Smyrnium olusatrum*, from western Algeria were characterized. Capillary GC-FID and GC/MS were used to investigate chemical composition of both essential oils, and the antifungal, antibacterial, molluscicidal and herbicidal activities were determined by % inhibition. Collective essential oil of *E. triquetrum* was dominated by falcarinol (74.8%) and octane (5.6%). The collective essential oil of *S. olusatrum* was dominated by furanoeremophilone (31.5%), furanodiene+curzurene (19.3%) and (E)-β-caryophyllene (11%). The *E. triquetrum* oil was tested and a pure falcarinol (99%) showed virtuous herbicidal and antibacterial activities against potato blackleg disease, *Pectobacterium atrosepticum*, and Gram-negative soil bacterium, *Pseudomonas cichorii* (85 and 100% inhibition, respectively), and high ecotoxic activity against brine shrimp, *Artemia salina*, and the freshwater snail, *Biomphalaria glabrata*, with an IC_50_ of 0.35 µg/mL and 0.61 µg/mL, respectively. Essential oil of *S. olusatrum* showed interesting antibacterial and ecotoxic activity and good herbicidal activity against watercress seeds, *Lepidium sativum* (74% inhibition of photosynthesis, 80% mortality on growth test on model watercress), while the furanoeremophilone isolated from the oil (99% pure) showed moderate herbicidal activity. Both oils showed excellent antifungal activity against *Fusarium*. Both oils and especially falcarinol demonstrated good potential as new biocontrol agents in organic crop protection.

## 1. Introduction

Today, it is widely recognized that herbicides may pose a significant risk to human health and to the environment. This risk has led to an increased interest in alternative strategies which have led to the development of biodegradable compounds [1,2]. Aromatic plants are increasingly used for pest control in bioagriculture. The essential oils and volatile constituents extracted from these plants are widely used as new replacement agents for biological control against microbial strains and pests, due to their specificity of action, their biodegradable nature and their commercial applications [3,4,5].

The genus sea holly (*Eryngium*) belongs to the family Apiaceae (Umbelliferae) and contains about 317 species [6,7]. Some of them have traditionally been used for the treatment of various anti-inflammatory disorders and antinociceptive activity [6]. Secondary metabolites isolated from plants belonging to this genus have displayed important biological activities, including antitumor, antibacterial, molluscicide, parasiticide, antimicrobial, antifungal, cytotoxic and phototoxic [6,7,8,9,10]. Many studies deal with phytochemical constituents of the *Eryngium* genus, reporting terpenoids, polyacetylenes, triterpenoid, saponins, steroids and phenolics [6,7,11].

Choukzerk (*Eryngium triquetrum* Vahl) is a plant that is endemic to North Africa and widely distributed in all parts of Algeria along with other *Eryngium* species [6]. The species grows particularly well in rocky pastures. It is considered to be a sturdier species, which the local people call “*choukzerk*” [12].

In the genus *Eryngium*, falcarinol has been identified as a major component in the essential oil in previous studies [7,12]; it is a highly bioactive polyacetylene with remarkable cytotoxic activity. It displays anti-inflammatory, antibacterial and a very strong antifungal potential [12]. It also displays antiproliferative [13] and nematocidal activity against pine wilt nematode (*Bursaphelenchus xylophilus*) and root-knot nematode (*Meloidogyne incognita*) [14]. It is also a natural pesticide and an anticancer agent [15].

Alexander (*Smyrnium olusatrum* L.) is certainly worthy of consideration. This plant, probably used since prehistoric times, became very popular in the time of Alexander the Great (4th century BC) and was widely cultivated and eaten by the Romans, who introduced it into Western and Central Europe, including the British Isles where it is now completely naturalized [16]. *S. olusatrum*, known as wild celery, is a biennial plant with a celery-like flavor that was used for several centuries as a vegetable and was then abandoned after the introduction of celery [16]. It was cultivated in Roman times for culinary and healing purposes [17]. The composition of the essential oils from *S. olusatrum’s* leaves, flowers and fruits is characterized by a high proportion of furanosesquiterpenes [18,19,20,21]. Roots and herbs are characterized by their high content of oxygenated sesquiterpenoids, most of which are furanosesquiterpenoids, while green and ripe fruits are dominated by hydrocarbons. In the roots and grass, the furanoeremophilone is the second most important furanosesquiterpenoid [22]. Furanodiene is a thermosensitive molecule that is subjected to a Cope rearrangement which yields the less active curzurene isomer. This rearrangement occurs during the heating of the plant material [23]. 

Essential oil of *S. olusatrum* shows natural antioxidant and herbicidal action against *Convolvulus arvensis* [24], high efficacy against larvae of the southern house mosquito (*Filariasis* vector *Culex quinquefasciatus*) with LD_50_ = 17.5 μg/mL [25] and cytotoxic activity on human colon cancer [16,21]. *S. olusatrum* oil is rich in furanoermophilone, which is an active molecule against fungi such as *Aspergillus*, *Trichoderma* and *Penicillium* [26]. This oil is also rich in curzurene, which has anticancer activity [27], and displays toxicity against mosquito species such as Celia mosquito (*Anopheles subpictus*), Asian tiger mosquito (*Aedes albopictus*) and Culex mosquito (*Culex tritaeniorhynchus*) [28].

In our characterization of Algerian aromatic plants in the field of crop protection [1,3,4,5,7,12,29,30], we reported here the chemical composition of the essential oils of *E. triquetrum* and *S. olusatrum* from western Algeria using a combination of flash chromatography followed by capillary GC(*RI*) and GC/MS. Biological activities were established through the characterization of phytotoxicity assays (germination, thylakoid activity and growth assay on the model grass watercress (*Lepidium sativum*)), ecotoxicology tests against *Artemia* larvae (*A. salina*) and freshwater snail (*B. glabrata*), antibacterial activity against potato blackleg disease (*Pectobacterium atrosepticum)* and Gram-negative soil bacterium (*Pseudomonas cichorii*) and finally antifungal activity against noble rot fungus (*Botrytis cinerea*) IP 1854.89 and wheat head blight fungus (*Fusarium graminearum*). 

## 2. Results

### 2.1. Composition of E. triquetrum Essential Oils

High-availability species have been identified as a result of the sampling of plant material in northwestern Algeria. Hydrodistillation of the fresh aerial parts of plants from Algeria resulted in (*w*/*w*) 0.02 to 0.11% (average 0.05%) of red essential oils. Table 1 presents the yields for all essential oils analyzed. The oils were first analyzed by both GC-FID and GC/MS and were primarily found to contain linear oxygenates (92.6%), dominated by falcarinol (74.8%), followed by octanal (5.6%) and nonanal (0.8%). Twenty components were identified by comparing their retention indices and their mass spectra with those from a “homemade” mass spectral library. Major compounds are schematized in Table 2.

### 2.2. Composition of S. olusatrum Essential Oils

The essential oil composition of *S. olusatrum* roots is reported in Table 1. A total of 41 components were identified, corresponding to 84.4% of the total oil. The main fraction consisted of furanosesquiterpenoid (59.1%). Collective essential oil was dominated by furanoermophilone (31.5%), furanodiene+curzurene (19.3%) and E-β-caryophyllene (11%). All of the furanosesquiterpenoids detected in this study are known to be precursors of sesquiterpene lactones, which are considered to be marker compounds of the genus *Smyrnium*. Furanoermophilone and furanodiene+curzurene are bioactive components. It is interesting to note that furanodiene is a thermosensitive molecule that is subjected to a Cope rearrangement to produce curzurene isomer. For this reason, we prefer to give the percentage of both molecules together (Figure 1). Major compounds are schematized in Table 3.

### 2.3. Germination

The germination is a preliminary bioherbicide test made in vitro (Petri boxes), giving the inhibitory effect of our samples on the germination and the growth of roots and stems on a model grass cress during a week. In all tests, *E. triquetrum* and *S. olusatrum* essential oils and pure falcarinol and furanoermophilone were tested. The germination test showed that both essential oils and major compounds had low inhibition of the germination rate and moderate inhibition of growth (Table 4).

### 2.4. Thylakoid Activity

The thylakoid test is a bioherbicide test which involves testing the inhibitory effect of our samples on photosynthesis. We used an atrazine positive control; this is the most common chemical molecule used as a herbicide in crop protection which has an inhibitory effect on the photosynthesis system.

As summarized in Table 5, essential oils of *S. olusatrum* and *E. triquetrum* and major compounds (falcarinol and furanoermophilone) showed photosynthesis inhibition activity of 74% and 43% for essential oils and 28% and 51% for furanoermophilone and falcarinol, respectively, at 0.01 mg/mL, with the inhibition of the positive control (atrazine) of 48% at the same concentration.

Compared with the positive control, the essential oil of *S. olusatrum*, especially, showed very interesting activity compared to the atrazine. Both falcarinol and *E. triquetrum* oil also showed very interesting results. Based on these results, falcarinol, *E. triquetrum* oil and especially *S. olusatrum* oil can be suggested as a substitution for chemical atrazine in crop protection to inhibit photosynthesis of weeds.

### 2.5. Growth Assay on the Model Grass Watercress (Lepidium sativum)

In order to confirm the potential ability to inhibit the photosynthesis of weeds, the growth test on the cress model was performed and results are presented in Figure 2. The experiments were carried out several times and in triplicate each time. We present the results after 7 days of treatment (7d) and after 15 days (15d). We took the photos using a camera. The plants after 7 days (7d) of treatment still kept their shape by comparing with the negative control but after 15 days (15d) the plants lost their shape, especially with the treatment with the essential oil of *E. triquetrum*, falcarinol and furanoermophilone.

This in vivo test is similar to the reality and shows the essential oil inhibition on the vegetative stage development of the plant. The observations were made after 7 and 14 days, and therefore the essential oils of *E. triquetrum* and *S. olusatrum* and major compounds showed good inhibition after 7 days, with a rate of mortality of 52% and 62%, respectively, and after 14 days, a mortality rate of 70% and 80%, respectively. For falcarinol and furanoermophilone, 68% and 50% rate of mortality was observed after 7 days and 88% and 67% after 14 days.

The results showed that both treatments (germination and growth inhibition) with the essential oils of *E. triquetrum* and *S. olusatrum* and major compounds demonstrated a weak inhibitory effect of sprouting on Petri dishes at a concentration of 0.1 mg/mL, and no large difference between both treatments was observed.

However, both treatments showed an inhibitory effect of photosynthesis and treatment with *S. olusatrum* oil especially, and falcarinol demonstrated an inhibition of photosynthesis of interest to 0.01 mg/mL compared with the second treatment. The results of the treatment on the pots showed that both treatments had an inhibitory effect on the growth of the model plants, and that the treatment with *S. olusatrum* and falcarinol showed an interesting mortality effect and growth inhibition after 14 days of treatment. Therefore, *S. olusatrum* oil and falcarinol can be considered to be a natural herbicide treatment against weeds.

### 2.6. Ecotoxicity Activity

The ecotoxicology test allows the evaluation of the toxicity of essential oils on aquatic fauna. Ecotoxicology test of *E. triquetrum* and *S. olusatrum* essential oils and their major compounds (falcarinol and furanoermophilone) was performed against *A. salina* and *B. glabrata*.

Figure 3 shows the mortality rate of *Artemia salina* larvae as a function of different concentrations of our samples. The experiment was carried out several times in triplicate, at different concentrations, and the results were determined after 24 h and 48 h by binocular observation.

For *A. salina*, results showed that the falcarinol and the collective essential oil of *E. triquetrum* have high activities with an IC_50_ of 0.35 µg/mL and 0.7 µg/mL, respectively. On the other hand, the essential oil of *S. olusatrum* and furanoermophilone showed moderate activity against *A. saline* with an IC_50_ of 5 µg/mL for furanoermophilone and 25 µg/mL for *S. olusatrum.*

Figure 4 shows the mortality of *B. glabrata* at different concentrations of our samples. The snails were fed fresh lettuce during the observation period (48 h). Each snail was observed under a binocular microscope to assess heart movements.

For *B. glabrata*, the results showed that falcarinol and *E. triquetrum* also presented very efficient activity with IC_50_ of 5.1 µg/mL and 0.5 µg/mL, respectively. Surprisingly, *S. olusatrum* showed low efficiency effect against *B. glabrata*, reaching 8.3% mortality at 50 µg/mL.

Our results provide evidence that: (i) the ecotoxicological effect may vary between the target organisms; and (ii) the ecotoxicological action of the two plants studied goes beyond the action of their major component [8,31]. Together, these results are very interesting, especially with falcarinol and the oil of *E. triquetrum*, which can be used to support control strategies of parasitic diseases related to limnic environments.

### 2.7. Antibacterial Activity against Phytopathogens

Antibacterial activity is carried out on phytopathogens, including *Pectobacterium atrosepticum*, responsible for the black leg of the potato and *Pseudomonas cichorii*, responsible for the disease of lettuce, celery and chrysanthemum. The antibiotic chloramphenicol is used as a positive control.

The results of the antibacterial activity of the essential oils are shown in Table 6. The results showed that falcarinol and *E. Triquetrum* oil have good antibacterial activity against *Pectobacterium* and *Pseudomonas* at 40 mg/mL (85% to 100% of inhibition after 24 h, respectively). On the other hand, the essential oil of furanoermophilone and *S. olusatrum* showed moderate activity against both strains (58% to 70%, respectively).

Based on these results, falcarinol can be a good natural alternative to chemical antibiotics used against *Pectobacterium* and *Pseudomonas*, especially in organic farming.

### 2.8. Antifungal Activity of Essential Oils on Mycelial Growth

Antifungal activity was tested on four fungal strains, including *Botrytis cinerea*, responsible for gray rot (tomatoes, strawberries), *Penicillium expansum*, responsible for fruit rot (apple/pear), *Zymoseptoria tritici*, responsible for septoria blight and *Fusarium graminearum*, responsible for cereal fusarium. A fungicide that was used as a positive control is Tebuconazol.

Antifungal activity of essential oils on mycelial growth is presented in Table 7. Falcarinol and *E. triquetrum* essential oil showed excellent activity against *Fusarium graminearum*, with lack of growth, moderate inhibition against *Botrytis cinerea* (55% and 43%), *Penicillium expansum* (33% and 27%) and *Zymoseptoria tritici* (17% and 34%) at very low concentration of 1.42*10^−1^ µL/mL air. Furanoermophilone and *S. olusatrum* showed more interesting inhibition activities than falcarinol and *E. triquetrum*, also with excellent activity against *Fusarium graminearum* with lack of growth, interesting activity against *Zymoseptoria tritici* (70% and 83%) and moderate activity against *Penicillium expansum* (30% and 39%) at very low concentration of 1.42*10^−1^ µL/mL air.

These results demonstrate that both oils and major compounds have antifungal potential against several fungi, especially *Fusarium graminearum*, and can be considered to be potential natural antifungal agents.

### 2.9. Fumigant Antifungal Activity 

The fumigant antifungal activity is tested on four strains, same as above. The difference between the two methods is that in the previous one, the strains are in direct contact with the essential oils and major compounds. Fumigant antifungal activity of essential oils is presented in Table 8.

Falcarinol and *E. triquetrum* essential oil showed excellent activity against *Fusarium graminearum* with lack of growth, moderate inhibition again *Botrytis cinerea* (55% and 43%), *Zymoseptoria tritici* (46% and 33%) and *Penicillium expansum* (17% and 8%) and at very low concentration of 1.42*10^−1^ µL/mL air. Furanoermophilone and *S. olusatrum* showed less activity than falcarinol and *E. triquetrum*, also with excellent activity against *Fusarium graminearum* with lack of growth, interesting activity against *Botrytis cinerea* (42% and 48%) and no activity against *Zymoseptoria tritici* and *Penicillium expansum*, also at very low concentration of 1.42*10^−1^ µL/mL air.

## 3. Discussion

Chemical compositions were performed by GC-FID and GC/MS-IE. Compositions were characterized by some interesting compound structures, such as falcarinol and furanoeremophilone’s skeleton. These molecules and essential oils demonstrated moderate activity in the preliminary bioherbicide test made in vitro, giving the inhibitory effect of our samples on the germination and the growth of roots and stems on model grass cress.

Bioherbicide assay, which involves testing the inhibitory effect of our samples on photosynthesis (thylakoid activity), showed very promising activity compared to atrazine positive control, the most common chemical molecule used as herbicide in crop protection which has an inhibitory effect on the photosynthesis system. Major compound of *Eryngium triquetrum*, falcarinol, showed photosynthesis inhibition activity of 74% at 0.01 mg/mL, with inhibition of the positive control (atrazine) of 48% at the same concentration. This result suggests the possibility of using falcarinol as potential industrial bioherbicide in inhibition of photosynthesis. This result was supported by the growth assay on the model grass watercress as evidence of potential ability to inhibit the photosynthesis of weeds. *S. olusatrum* oil and falcarinol demonstrated an inhibition of photosynthesis of interest to 0.01 mg/mL.

The ecotoxicology effect of essential oils and major compounds tested against *A. salina* and *B. glabrata* demonstrated a very strong effect at low concentration, especially of falcarinol and E. triquetrum oil, which can be suggested in support control strategies of parasitic diseases related to limnic environments.

Finally, antibacterial and antifungal activities were carried out against phytopathogens responsible for many diseases, as black leg and lettuce and mycelial growth were tested against gray rot, fruit rot, septoria blight and cereal *Fusarium*. All these activities showed interesting results, offering very large spectra of use of natural compound as potential agents in organic crop protection.

The plants of the family Apiaceae are known for the richness of bioactive compounds in their essential oils. In this study, we investigated the herbicidal and pesticidal potency of two Apiaceae species, as alternatives to synthetic chemical pesticides. *E. triquetrum* is a species rich in terpene alcohol, falcarinol, and the second species, *S. olusatrum*, is rich in furanosesquiterpenes molecules. This study showed that the essential oil of *E. triquetrum* had strong herbicidal, ecotoxic and antibacterial activity, and that this high activity was due to the presence of falcarinol, which is a compound associated with various biological activities. The second essential oil in this study, *S. olusatrum* essential oil, also showed good herbicidal activity, yet its ecotoxic and antibacterial activity was less important when compared to that of *E. triquetrum.*

Weed control is the botanical component of pest control, which attempts to stop weeds, especially noxious weeds, from competing with desired flora and fauna, including domesticated plants and livestock, and in natural settings, preventing non-native species from competing with native species. Weed control in conventional and organic agriculture is a major issue. In organic farming, it becomes impossible to use chemical herbicides, which opens the way to the search for natural molecules with bioherbicidal power. The work in progress allows us to offer a perspective for the use of natural compounds, such as *E. triquetrum* and *S. olusatrum* essential oils and falcarinol in organic weed control and crop protection. In the same way, the use of falcarinol may also be considered as a good alternative to atrazine in conventional agriculture, since falcarinol has shown better results than atrazine, considered to be the most widely used synthetic herbicide in the world. This may also be helped by the problems with atrazine and its current ban by several countries due to its contamination of water sources.

The presence of falcarinol at very high percentages in *Eryngium* (93% for some harvesting stations) brings the perspective of considering this essential oil as a possible source of this molecule since it is the only source described in the literature that presents falcarinol at such high rates. The good results of falcarinol’s activities as a herbicide, antibacterial against *Pseudomonas* and *Pectobacterial* and antifungal against *Fusarium* at very low concentrations, therefore allows us to consider its use on an industrial scale for crop protection.

The essential oil of *S. olusatrum* has shown more interesting herbicidal activity, also better than furanoermophilone. Our work has shown that this activity is not only due to the major compound (furanoermophilone), which was tested as a pure molecule and did not give similar results as the essential oil, but that it is rather a synergistic effect between the different molecules of this oil that is responsible for this activity. In addition to the herbicidal activity, this oil has also shown very good results as antifungal activity against *Zymoseptoria* and *Fusarium*, which gives it a wide spectrum of use as a bioherbicide, biofungicide and also antibacterial agent in organic farming and crop protection.

In addition, both compounds gave interesting results which could be useful to limit transmission of water-borne parasitic diseases in endemic areas. In this sense, the World Health Organization (WHO) via “WHO Strategic and Technical Advisory Group for Neglected Tropical Diseases (NTD)” stimulates strategies to control parasitic diseases through ecological control of population of the intermediate host, such as *B. glabrata* snails for schistosomiasis control. Both compounds studied here presented molluscicide properties, even in different levels, and should be considered instead of deleterious chemical pesticides, such as Roundup (glyphosate) [31]. In the past decades, many plants were tested as a source of potential phytochemical molluscicides [32,33,34], and *E. triquetrum* and its major component (falcarinol) are promising as molluscicides against the fresh snail *B. glabrata*, meeting the requirements of WHO for use as a natural molluscicide [35].

The two essential oils tested in this work reinforce the willingness to use natural substances as healthy control substances against pests affecting the development of organic agriculture and crop protection worldwide.

## 4. Materials and Methods

### 4.1. Plant Material and Oil Isolation

The aerial parts of *E. triquetrum* and root parts of *S. olusatrum* were harvested from May to June 2017 and December 2016, respectively, for each species, during the vegetative cycle of the plant, in northwestern Algeria (Tlemcen). Voucher specimens were deposited in the herbarium of the University of Tlemcen (voucher numbers of each specimen analyzed are ET-1011-S1 and SO-1012-S1, respectively, for *E. triquetrum* and *S.*
*olusatrum*). The essential oils were obtained from the aerial parts and fresh roots of both species, through a 4 h hydrodistillation process with a Clevenger-type device, which met the European Pharmacopoeia standards [36] and yielded (*w*/*w*) from 0.035 to 0.078% of *E. triquetrum* and 0.004 to 0.009% of *S. olusatrum*.

### 4.2. Gas Chromatography

Essential oil samples were analyzed for each individual plant with a Thermo Scientific Focus GC with a flame ionization detector (FID) using a DB-5 fused-silica capillary column (30 m × 0.25 mm × 0.25 μm) and SPB-1 with helium as the carrier gas (flow 1.0 mL min^−1^) and an injection volume of 1 μL (2% solution in cyclohexane); the initial pressure was 1.0 Pa (Thermo Electron SAS, Villebon-sur-Yvette, France) The column temperature was programmed as follows: 60 °C for 3 min, 60–240 °C (3 °C min^−1^), 240–300 °C (10 °C min^−1^) and 300 °C for 10 min. The injector temperature was 250 °C, with injection in the “splitless” mode. The analyses lasted 80 min. Samples were analyzed in triplicate. The retention indices (RIs) of the compounds were determined to the retention times (RT) for a series of n-alkanes (C7–C40): commercial solution, using the equation of Van den Dool and Kratz.

### 4.3. Gas Chromatography-Mass Spectrometry

The essential oils were also analyzed and identified using mass spectrometry in chromatography coupled to mass selective detector DSQ II with the same conditions as above. The condition had impact energy of 70 eV. Qualitative analysis of compounds was based on the comparison of their spectral mass and their relative retention time with those of NIST mass spectra database and Kovats RI on the chromatograms recorded in full scan or in SIM mode using the characteristic ions.

### 4.4. Compound Identification

Identification of the components was based on a comparison of their GC retention indices (RI) with data found within the literature [37,38], and/or from MS databases [39].

### 4.5. Oil Fractionation

Collective oils of *E. triquetrum* and *S. olusatrum* were obtained from a mixture of all individual essential oils prepared from plant material harvested in west Algerian locations. For *E. triquetrum*, collective oil was fractioned and falcarinol was purified (99%). Essential oil (105 mg) was separated on a chromatographic column using silica gel and cyclohexane-diethylether as eluent. Falcarinol was purified by thin layer chromatography (TLC) using KMnO_4_ as revelator. A mass of 65.7 mg of yellow colored oil was obtained at 99% purity (determined by GC analysis). In order to know the configuration of falcarinol, the measurement of the rotatory power is necessary. Optical rotation of falcarinol was measured in chloroform, using a polarimeter model ATAGO, POLAX-2L (Ficher Bioblock Scientific, Illkirch, France) The test consisted of measuring the rotatory power of the oil at 0.5% and the purified falcarinol at 0.5%. α_D (oil)_= +30°, α_D (falcarinol)_= +46°.

*S. olusatrum* collective oil was fractionated. A 391 mg portion of the latter was fractionated on a conventional silica gel column. The eluting solvents were n-hexane (A) and ethyl acetate (B), which yielded several fractions (a-i) with an elution gradient: a (A: 100%; B: 0%), b (A: 98%, B: 2%), c (A: 97%, B: 3%), d (A: 96%, B: 4%), e (A: 95%; B: 5%), f (A: 90%, B: 10%), g (A: 85%, B: 15%), h (A: 50%, B: 50%), i (A: 0 % B: 100%).

### 4.6. Phytotoxicity Assays

#### 4.6.1. Germination

Determination effects on pregermination were performed on watercress seeds (*Lepidium sativum*) on cellulose disks in Petri dishes according to [40] with some modifications. Ten seeds were placed on each disk and soaked in 2 mL of solution containing the essential oil. A final concentration of 0.1 and 0.01 mg/mL of essential oil in 1% DMSO was used. Four replicates were performed for each concentration. The effect of the treatments on germination and growth rate was evaluated through periodic counting (every 24 h) of the number of seeds that sprouted, as well as the length of the stems and roots.

#### 4.6.2. Thylakoid Activity

##### Thylakoid Membrane Preparation

Thylakoid membranes were isolated from spinach leaves (*Spinacia oleracea* L.), according to methods reported by Rouillon et al. [41]. First, 100 g of fresh leaves was washed with distilled water and homogenized in 300 mL of extract buffer. The homogenate was filtered through sieves with a pore diameter of 100 µm, and then centrifuged for 2 min at 3300 RPM. Supernatant was eliminated and the pellet was resuspended in the flushing solution and centrifuged for 1 min at 3300 RPM. Pellet was resuspended in 1 mL of distilled water to disrupt chloroplasts. The obtained thylakoid membranes were stirred for 10 s. Thylakoid membranes were then placed in the flushing solution and centrifuged for 2 min at 3300 RPM. The pellet was resuspended in the immobilization buffer. The concentration of chlorophyll was estimated. The preparation was diluted to 2 mg chlo/mL by immobilization (manitol, 3-morpholinopropane-1-sulfonic acid, ethylene diamine tetra-acetic acid and bovine serum albumin) buffer and stored in liquid nitrogen.

##### Determination of Thylakoid Membrane Activity

Thylakoid membrane activity was estimated, according to the method reported by [42], with several modifications. Thylakoid membranes were diluted to 30 µg chlo/mL with the immobilization buffer. Samples containing 30 µL of thylakoid membrane solution were mixed with 20 µL of DMSO 37.5% (control) or herbicide solution (atrazine 100 µM) or with an essential oil of 0.1, 0.05, 0.01 mg/mL, 100 µL 0.3 mM DPIP in 0.1 M MOPS (3-morpholinopropane-1-sulfonic acid), at a pH of 6.5 in the micro titer plate wells. Thylakoid membranes were prepared earlier and stored in micro titer plates in the dark at 4 °C. The samples were illuminated for 20 min with an OSRAM L 36 W/ 865; SYLVANIA F 36 W/ GRO–T8 lamp and absorbance was measured at 530 nm using reader Thermo Scientific Varioskan LUX. The initial and final absorbencies for each well were recorded. The activity of thylakoid membrane preparations was calculated from the data on the amount of reduced DPIP, the concentration of chlorophyll and the time of illumination, and was expressed as µmol DPIP/mg/chl/h.

#### 4.6.3. Growth Assay on the Model Grass Watercress (*Lepidium sativum*)

For optimal seed emergence growth, the following proper proportion of compost and soil must be mixed: 1/3 compost and 2/3 soil. The soil was put in pots (40 g), and to keep the soil moist, 15 mL of tap water was added to each box. Rows of *L. sativum* seeds were added to each pot, with 3 seeds per row and a total of 3 rows (9 seeds) per pot. Essential oils at a concentration of 82.8 mg/100 mL of water with 0.35% DMSO were used as the test and control compounds. Sulcotrine served as a positive control, while water with 0.35% DMSO served as a negative control. Next, 1 mL of each solution was sprayed in each of the pots. The assay was carried out in triplicate. Observations were made daily.

#### 4.6.4. Ecotoxicology Test on *Artemia larvae* (*Artemia salina*)

##### Harvesting Larvae of *Artemia* (48 h before the Test)

*Artemia salina* larvae were lyophilized in a Petri dish containing seawater prepared from synthetic sea salt (3.5 g in 100 mL of distilled water). A foil-covered lid was placed on half of the box which was then positioned under a lamp for 48 h. After hatching, the larvae naturally moved to the lit area of the box.

##### Assay

The test was carried out in 96-well micro plates according to the method reported by [43]. A preliminary test was performed with a wide range of concentrations (100, 500 and 1000 ppm) to determine the optimal concentration range (results not shown). Following this preliminary test, a secondary test was performed with higher concentrations (5, 2.5, 2, 1.5, 1, 0.5 mg/mL) of each oil in order to estimate the IC_50_. Each concentration was tested in triplicate. In each well, we placed 2 μL of oil, 198 μL of seawater and 10 shrimp. Controls were made by depositing 100% seawater containing 1% solubility solvent (DMSO). After 24 and 48 h, the number of dead shrimp was determined through the binocular observation of shrimp mobility.

#### 4.6.5. Ecotoxicological Test against *Biomphalaria glabrata*

##### Harvesting the Freshwater Snail *B. glabrata*

*B. glabrata* originating from Recife Brazil (BgBRE2) was used for this study. These snails were maintained in rearing chambers at 26 °C, 12/12 h light/dark period and fed ad libidum with fresh lettuce. A Puerto Rican strain (NMRI) of *S. mansoni* was used in this study to determine the effects of molluscicide on the susceptibility of infected snails and host–parasite compatibility.

##### Molluscicidal Bioassays

Biological assays against the snail *B. glabrata* were carried out with 12 snails with shell diameters from 6–8 mm individually exposed to different concentrations of essential oil or DMSO used as control. All snails were immersed for 24 h and then washed and transferred to tanks filled with drilling water. Snails were fed with fresh lettuce ad libidum during the observation period (48 h). Each snail was observed under binocular microscope to appreciate the heart movements.

### 4.7. Screening for Antibacterial Activity

#### 4.7.1. Bacterial Strain

Antimicrobial activity was tested in vitro against *Pectobacterium atrosepticum* (strain CFBP-5384) and *Pseudomonas cichorii* (strain CFBP 4407) provided by the Stock Culture Collections of Phytopathogenic Bacteria (CFBP, INRA Angers, France). Bacteria were maintained for extended periods as deep-frozen cultures (−20 °C) and cultivated on King B Agar Medium at 25 °C for 48 h. Strain was grown in peptone water liquid nutritive medium (WPN) at 25 °C overnight for use in the antibacterial activity test.

#### 4.7.2. Antibacterial Activity against Phytopathogens

The microplate bioassay (microdilution) was used to study the antimicrobial activities of *P. atrosepticum* and *P. Cichorii*, according to the method reported by Meziani et al. [44]. The essential oil dissolved in aqueous dimethylsulfoxide (DMSO) at 40, 20, 10, 5, 2.5, 1.25, 0.625 and 0.312 mg/mL concentration was pippeted into sterile 96-well plates based on a National Committee for Clinical Laboratory Standard method (NCCLS).

The bacterial suspension was adjusted with sterile distilled water to 2.105 cfu/mL concentration. In brief, 96-well plates were prepared by dispensing 198 µL of microorganism suspension into each well. Then, 2 µL from their serial dilutions of extracts was transferred into five consecutive wells. The last well containing 198 µL of WPN medium and 2 µL of the serial dilutions of extracts was used as a negative control. The final volume in each well was 200 µL. The plate was covered with a sterile plate sealer and then incubated at 25 °C for 48 h. After agitation, microorganism growth was estimated by reading absorbance in microplate wells at 600 nm with a Microplate Reader Thermo Scientific Varioskan LUX (Life Technologies SAS, Villebon-sur-Yvette, France). We used chloramphenicol at a concentration of 10 μg/mL as a positive control. Inhibition of bacterial growth was measured after 24 h of incubation at 23 °C using a digital caliper which made it possible to calculate growth inhibition with the following formula:% inhibition = [(DO_control_−DO_extrait_)/DO_control_] × 100

### 4.8. In Vitro Antifungal Activity of Selected Extracts on Mycelial Growth

#### 4.8.1. Fungal Strains

*Botrytis cinerea*, *Fusarium graminearum*, *Zymoseptoria tritici* and *Penicillium expansum* (IP 1854.89; MAT REF 04-14-02-02; CBS 398.52 and IP 1350.82) were grown on malt agar medium and incubated at 23 °C in darkness.

#### 4.8.2. Antifungal Activity by Contact Method

The essential oils were evaluated for antifungal activity based on their ability to inhibit mycelial growth [45]. First, 10 μL of essential oils was put on a cellulose disk. The latter was placed in the middle of the agar in a Petri dish. A 3 mm diameter mycelial buffer of each pathogen from a 7-day-old culture was placed in the center of the medium. The negative control (100% medium and cellulose disk without essential oil) and the positive control (a 10 ppm solution of tebuconazole (Sigma-Aldrich, St. Louis, Missouri, USA)) were prepared in the same way. Each treatment was applied in triplicate. The radial growth of the mycelium was measured after 1 to 6 days of incubation at 23 °C using a digital caliper which allowed calculating the index with the following formula:Antifungal index (%) = (1 − Da/Db) × 100
where Da is the diameter of growth zone in the experimental plate (cm) and Db is the diameter of growth zone in the control plate (cm).

#### 4.8.3. Fumigant Antifungal Activity

Essential oils were evaluated for fumigant antifungal activity based on their ability to inhibit mycelial growth, according to Benomari et al. [4]. Six-millimeter-diameter mycelial plugs for each fungal strain from a 7-day-old culture were placed into a Petri dish. Essential oils were introduced onto a 6 mm cellulose disk and placed on the agar-free lid of the Petri dish. The antifungal strains used in this method are the same as those used in the previous method (antifungal activity by contact method). A negative control (cellulose disk without essential oil) was performed in the same way. Petri dishes were then sealed with parafilm and incubated at 23 °C in the dark. Mycelial radial growth was measured after 3 to 7 days of incubation, and the antifungal index was calculated with the formula as follows:Antifungal index (%) = (1 − Da/Db) × 100

## 5. Conclusions

The present study describes the chemical composition and herbicidal, antifungal, antibacterial and molluscicidal activities of two Algerian essential oils, *E. triquetrum* oil, which is dominated by falcarinol (a C-17 polyacetylene with interesting biological activities) and *S. olusatrum* oil, which is rich in furanosesquiterpenes (mainly furanoermophilone).

Capillary GC-FID and GC/MS-IE were used to investigate chemical composition of both essential oils. Collective essential oil of *E. triquetrum* was dominated by falcarinol (74.8%) and octane (5.6%). Collective essential oil of *S. olusatrum* was dominated by furanoeremophilone (31.5%), furanodiene+curzurene (19.3%) and (E)-β-caryophyllene (11%).

Activity stemming from *E. triquetrum* essential oil, which has high falcarinol content, was of great interest, as falcarinol is responsible for many important biological activities such as antimicrobial activity as well as potentially interesting herbicidal and ecotoxic activities. *E. triquetrum* essential oil and isolated falcarinol are promising against fresh snail *B. glabrata*, main intermediate host of schistosomiasis. In addition, *E. triquetrum* oil and falcarinol have shown interesting results as an alternative molluscicidal agent to reduce snail infections with schistosomes in transmission foci. Future research could investigate the biological activity of synthetic or natural falcarinol and its derivatives to develop new agents to prevent schistosomiasis transmission.

The *E. triquetrum* oil tested and a pure falcarinol (99%) showed virtuous herbicidal and antibacterial activities against potato blackleg disease, *Pectobacterium atrosepticum*, and Gram-negative soil bacterium, *Pseudomonas cichorii* (85 and 100% inhibition, respectively), and high ecotoxic activity against brine shrimp, *Artemia salina*, with an IC_50_ of 0.35 µg/mL. In the same case, essential oil of *S. olusatrum* showed interesting antibacterial and ecotoxic activity and good herbicidal activity against watercress seeds, *Lepidium sativum* (74% inhibition of photosynthesis, 80% mortality on growth test on model watercress), while the furanoeremophilone isolated from the oil (99% pure) showed moderate herbicidal activity.

Both essential oils showed excellent antifungal activity, especially against *Fusarium*. Both *E. triquetrum* and *S. olusatrum* essential oils and their major compounds could be considered as biocontrol agents to protect trees from bacteria and may serve as important alternatives to chemical pesticides in crop protection.

## Figures and Tables

**Figure 1 antibiotics-10-00636-f001:**
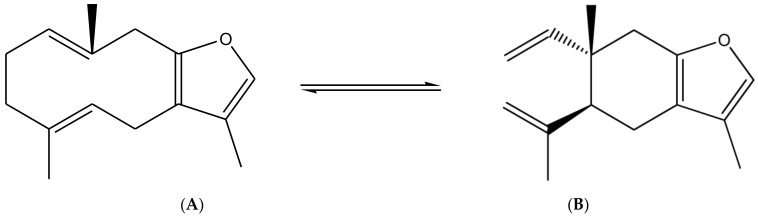
Cope rearrangement of furanodiene (**A**) to curzurene (**B**).

**Figure 2 antibiotics-10-00636-f002:**
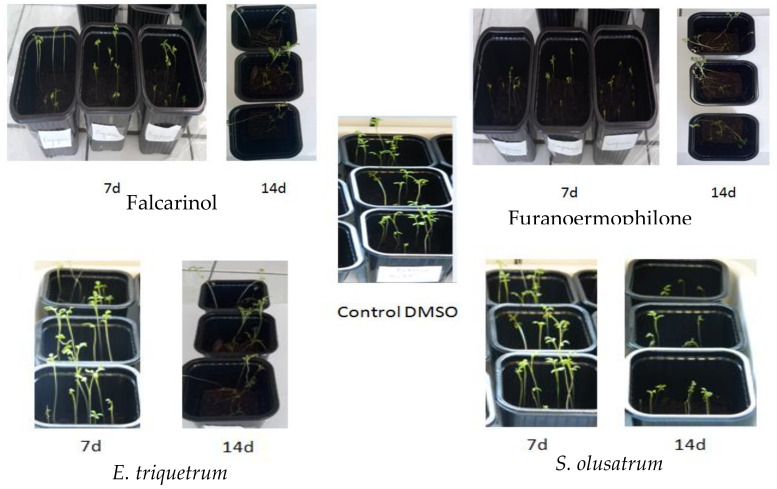
The effect of essential oils and major compounds on growth on model grass cress (*Lepidium sativum*) after 7 and 14 days.

**Figure 3 antibiotics-10-00636-f003:**
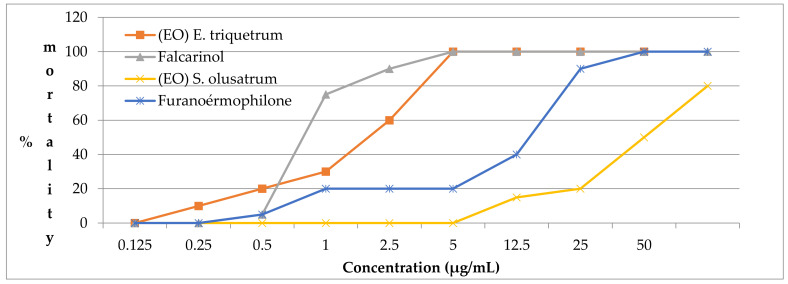
The death rate of *Artemia* at different concentrations of essential oils *E. triquetrum* and *S. olusatrum* and major compounds.

**Figure 4 antibiotics-10-00636-f004:**
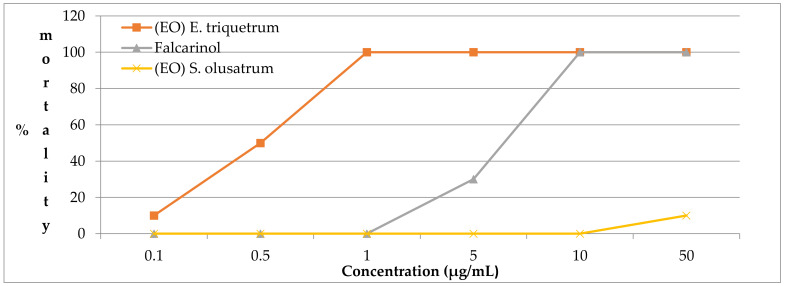
The death rate of *B. glabrata* at different concentrations of essential oils *E. triquetrum* and *S. olusatrum* and major compounds.

**Table 1 antibiotics-10-00636-t001:** Chemical composition of *S. olusatrum* and *E. triquetrum* essential oils from Algeria.

			*S. olusatrum*	*E. triquetrum*	
No	Components †	RI _a_‡	Roots §	Stems §	Identification
1	butanol	638	0.5		RI, MS
2	heptane	703		0.6	RI, MS
3	hexanal	774		0.5	RI, MS
4	heptanal	877		tr *	RI, MS
5	α-thujene	925	0.1		RI, MS
6	α-pinene	933	0.8		RI, MS
7	sabinene	968	0.1		RI, MS
8	β-pinene	974	3		RI, MS
9	**octanal**	979		**5.6**	RI, MS
10	myrcene	983	0.1		RI, MS
11	α-phellandrene	996	tr *		RI, MS
12	∆-3-carene	1008	2.2		RI, MS
13	p-cymene	1013	0.3		RI, MS
14	limonene	1022	0.2		RI, MS
15	E-β-ocimene	1037	0.1		RI, MS
16	(E)-2-octenal	1039		0.2	RI, MS
17	1-octanol	1063		tr *	RI, MS
18	nonan-2-one	1077		tr *	RI, MS
19	α-terpineol	1079	0.1		RI, MS
20	nonanal	1081		0.8	RI, MS
21	(E)-2-nonenal	1133		0.5	RI, MS
22	p-menth-8-en-1-ol	1139	0.1		RI, MS
23	(Z)-2-nonen-1-ol	1155		0.2	RI, MS
24	menthofurane	1156	0.1		RI, MS
25	p-cymen-8-ol	1160	0.1		RI, MS
26	α-terpineol	1172	Tr *		RI, MS
27	octanoic acid	1174		Tr *	RI, MS
28	1-decen-3-ol	1181		0.7	RI, MS
29	decanal	1183		Tr *	RI, MS
30	citronellol	1209	0.3		RI, MS
31	cuminaldehyde	1212	0.2		RI, MS
32	3-dodecen-1-yne	1214		0.5	RI, MS
33	carvone	1225		0.5	RI, MS
34	(E)-2-decanal	1251		Tr *	RI, MS
35	piperitenone	1286	Tr *		RI, MS
36	(E,E)-2,4-decadienal	1289		Tr *	RI, MS
37	citronellyl acetate	1333	0.1		RI, MS
38	α-copaene	1379	0.1		RI, MS
39	β-bourbonene	1382	0.2		RI, MS
40	β-elemene	1387	1.1		RI, MS
**41**	**E-β-caryophyllene**	1420	**11**		RI, MS
42	δ-elemene	1427	0.3		RI, MS
43	α-humulene	1450	0.9		RI, MS
44	β-ionone	1454		0.3	RI, MS
45	germacrene D	1475	0.1		RI, MS
46	**furanodiene**	1480	**19.1**		RI, MS
47	**curzurene**	1484	**0.2**		RI, MS
48	3,4-dimethyl-5-pentyl-5H-furan-2-one	1486		2.7	RI, MS, Ref **
49	α-bulnesene	1507	0.2		RI, MS
50	α-cadinene	1510	0.3		RI, MS
51	δ-cadinene	1515	0.5		RI, MS
52	γ-undecalactone	1524		tr *	RI, MS, Ref **
53	dodecanoic acid	1547		tr *	RI, MS
54	germacrene B	1552	1.5		RI, MS
56	caryphylleneoxyde	1572	2.2		RI, MS
57	isocaryophyllen-14-ol	1629	0.7		RI, MS
58	4-β-4-cadin-9-en-15-al	1677	0.9		RI, MS
59	14-hydroxy-δ-cadinene	1776	0.5		RI, MS
60	**Furanoeremophilone**	1860	**31.5**		RI, MS
61	hexadecanoic acid	1968		1.9	RI, MS
62	**Falcarinol**	2026		**74.8**	RI, MS
63	α-kaurene	2049	1.4	0.3	RI, MS
64	E-phytol	2106	2.3		RI, MS
65	Tridecane	2303	0.5		RI, MS
66	Tetradecane	2403	0.4		RI, MS
67	Pentadecane	2503	0.1		RI, MS
	**Total identification %**		**84.4**	**90.1**	
	EO, yields (%) (*w*/*w*)		0.004	0.02	
	Oxygenated compounds		39.6	89.2	
	Hydrogenated compounds		44.8	0.9	

† Order of elution is given on a polar column (DB5), ‡ retention indices on DB5 column (RIa), § collective oils: mixture of all Algerian *S. olusatrum* and *E. triquetrum* (roots and stems essential oils, respectively). * tr = trace (< 0.05%). RI: retention indices; MS: mass spectrometry in electronic impact mode. ** All compounds were identified by comparing their EI-MS and retention indices with references compiled in the in-house library, except for compounds **48** and **52**.

**Table 2 antibiotics-10-00636-t002:** Falcarinol structure and information.

Molecule	Name	Formula	IUPAC Name
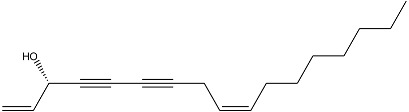	Falcarinol (74.8%)	C_17_H_24_O	(3S,9Z)-Heptadeca-1,9-diene-4,6-diyn-3-ol

**Table 3 antibiotics-10-00636-t003:** Furanoeremophilone and furanodiene+curzurene structures and information.

Molecule	Name	Formula	IUPAC Name
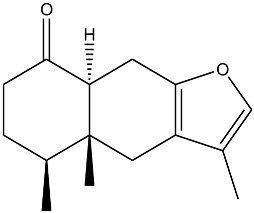	Furanoeremophilone (31.6%)	C_15_H_20_O_2_	3,4a,5-trimethyl-4a,5,6,7,8a,9-hexahydro-4H-naphtho[2,3-b]furan-8-one
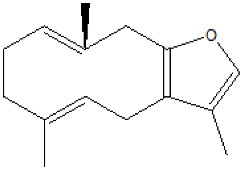	Furanodiene (19.1%)	C_15_H_20_O_2_	3,6,10-trimethyl-4,7,8,11-tetrahydro-cyclodeca[b]furan
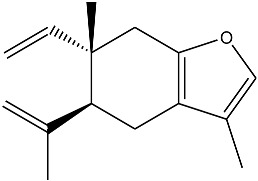	Curzurene (0.2%)	C_15_H_20_O	5-isopropenyl-3,6-dimethyl-6-vinyl-4,5,6,7-tetrahydro-benzofuran

**Table 4 antibiotics-10-00636-t004:** Germination and growth inhibition results of *E. triquetrum* and *S. olusatrum* essential oils and the major compounds.

EO/Molecules0.1 mg/mL	% Inhibition	
	Germination Rate	Growth Inhibition
*E. triquetrum*	3	12
Falcarinol	5	18
*S. olusatrum*	0	18
Furanoermophilone	0	10

**Table 5 antibiotics-10-00636-t005:** Thylakoid test results of *E. triquetrum* and *S. olusatrum* essential oils and the major compounds.

EO/Molecules 0.01 mg/mL	% Inhibition of Photosynthesis
*E. triquetrum*	43
Falcarinol	51
*S. olusatrum*	74
Furanoermophilone	28
Atrazine	48

**Table 6 antibiotics-10-00636-t006:** Antibacterial activity of essential oils and the major compounds.

EO/Molecules 40 mg/mL	% Inhibition after 24 h	
	*Pectobacterium*	*Pseudomonas*
*E. triquetrum*	85	100
Falcarinol	100	100
*S. olusatrum*	63	58
Furanoermophilone	70	70
Chloramphenicol	100	100

**Table 7 antibiotics-10-00636-t007:** Antifungal activity of essential oils and the major compounds on mycelial growth.

EO/Molecules 1.42*10^−1^ µL/mL Air	% Inhibition after 3 Days			
	*Botrytis Cinerea*	*Penicillium Expansum*	*Fusarium Graminearum*	*Zymoseptoria Tritici*
*E. triquetrum*	43	27	Growth	17
Falcarinol	55	33	Growth	34
*S. olusatrum*	48	39	Growth	83
Furanoermophilone	42	30	Growth	70
Tebuconazole	100	100	100	100

**Table 8 antibiotics-10-00636-t008:** Fumigant antifungal activity of *E. triquetrum* and *S. olusatrum* essential oils and the major compounds.

EO/Molecules 1.42*10^−1^ µL/mL Air	% Inhibition after 3 Days			
	*Botrytis Cinerea*	*Penicillium Expansum*	*Fusarium Graminearum*	*Zymoseptoria Tritici*
*E. triquetrum*	43	8	Growth	33
Falcarinol	55	17	Growth	46
*S. olusatrum*	48	0	Growth	0
Furanoermophilone	42	0	Growth	0
Tebuconazole	100	100	100	100

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
