# Peer review of "Essential Oils from Two Apiaceae Species as Potential Agents in Organic Crops Protection"

_antibiotics, 2021, doi:10.3390/antibiotics10060636_

Round 1
Reviewer 1 Report
The topic is interesting and worth researching. Namely, natural resources should be explored and used as much as possible.
I don't know if it is necessary to write the Latin names of plants in curly braces or can they fit into the text?
Line 381 can you specify the name of the author by whom you performed the method as in line 472?
Do you have a picture for 4.8.2. method? It is not clear at what distance were the disc with the essential oil and the planted mold.
Figures 3 and 4 need to be rearranged and an additional description is needed below the figure.
Fig 2. The images are not uniform and the plants are not well visible as well as the influence of the tested components on them. Either select larger images showing the effect and control it or display it differently.
Author Response
Dear reviewer,
Thank you for your excellent comments. The responses are enclosed in the attached file
Best regards

Reviewer 2 Report
Review comment on “Essential Oils from Two Apiaceae Species as Potential Agents in Organic Crops Protection” General comment.
This is a research article to discuss the application of herb extraction falcarinol and octane in bacterial inhibition and seeds germination effects. The paper is well organized and the result is very interesting. I recommend this manuscript, but the author should do a major revision. In addition, the authors may need more mass spectrometry training, and improving the research quality.
Comments 1. ” Table 1. Chemical composition of S. olusatrum and E. triquetrum essential oils from Algeria.” Did you run the blank with your target chemicals?
2. Same in the table, there are several same chemicals, please check the Nuclear mass ratio in your gc/ms map, if not, you can share the data, we can do together to make sure the results;
3. Please update a color figure, it is hard to clarify the changes;
4. Please describe more of the Thylakoid activity, as I know, thylakoid is a organism, I more prefer chlorophyll,
5. Line 214-217, I think you should add some references and cases for your result explanation; And all the result parts;
6. I am not sure the discussion is strong enough, please combine the discussion and result parts, and then the structure will be stronger; Also the discussion part does not match well with your results;
7. You can give one or two sentences to conclude your research.
Author Response

(The authors gave the same response as above.)

Round 2
Reviewer 1 Report
Although part of the text has been modified, I do not see that Figure 2 and 3 have been modified . An informative description should be written below the Figure. How many times the experiment was repeated and similar. Also Figure 2 is not modified and I don't see an explanation why.
Author Response
Dear reviewer,
As requested, I send you the revised paper in track changes.
Regards,
Yacine Boumghar
